# Continuous Perception Benchmark

## Abstract

Humans continuously perceive and process visual signals. However, current video models typically either sample key frames sparsely or divide videos into chunks and densely sample within each chunk. This approach stems from the fact that most existing video benchmarks can be addressed by analyzing key frames or aggregating information from separate chunks. We anticipate that the next generation of vision models will emulate human perception by processing visual input continuously and holistically. To facilitate the development of such models, we propose the Continuous Perception Benchmark, a video question answering task that cannot be solved by focusing solely on a few frames or by captioning small chunks and then summarizing using language models. Extensive experiments demonstrate that existing vision models, whether commercial or open-source, struggle with these tasks, indicating the need for new technical advancements in this direction.

## 1 Introduction

Video understanding is a foundational task in computer vision that has been extensively studied for decades. Over the years, a variety of methods have been developed, utilizing architectures that range from temporal convolutions [1] to 3D convolutions [2, 3] and, more recently, transformers [4, 5]. The current trend towards scaling has led to the emergence of multi-modal foundation models [6, 7, 8], which represent the state-of-the-art in video understanding [9, 10, 11, 12, 13, 14]. These models are trained on massive amounts of web data, demonstrating exceptional generalization capabilities across different tasks. Additionally, they can engage in open-vocabulary, multi-round interactions with users, a capability that previous specialized models lacked [9, 12, 14]. This advancement holds significant promise for real-world applications, such as personal assistants.

Despite the progress, current video foundation models process videos differently from humans. Typically, these models use one of two approaches. The first approach (top left of Figure 1) involves sparsely sampling frames from the input video and only processing those sampled frames [9, 10, 11, 12, 13, 14]. The second approach (top right of Figure 1) divides the input video into separate chunks, processes each chunk independently by captioning it, and then summarizes the entire video's information by using a large language model (LLM) to process the generated captions [15, 16, 17]. In contrast, humans perceive and process visual signals densely and continuously. We anticipate that the next generation of visual foundation models should mimic this human approach, processing input video comprehensively without resorting to sparse sampling or dividing it into chunks. Firstly, sparse sampling or chunk processing can result in the loss of global temporal information across the entire video. More importantly, we believe that the ability to continuously process visual signals efficiently is crucial for learning critical concepts such as compositionality [18], intuitive physics [19], and

Submitted to the 38th Conference on Neural Information Processing Systems (NeurIPS 2024) Track on Datasets and Benchmarks. Do not distribute.

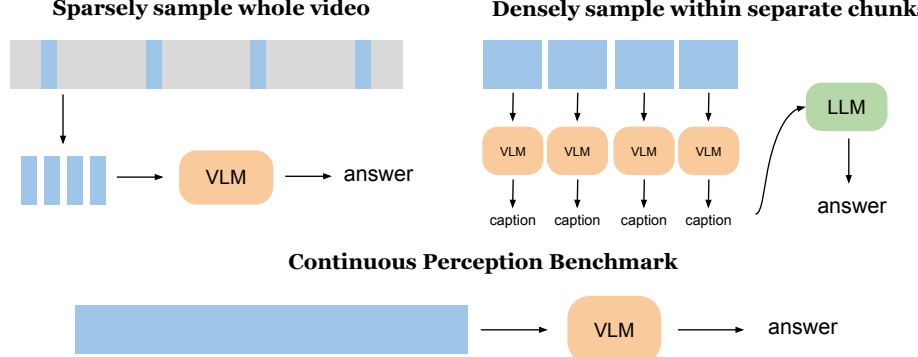

Figure 1: (Top) Existing video understanding models process videos in one of two ways: either by sparsely processing the entire video or by densely processing it in chunks. Similarly, most existing video benchmarks can be addressed using these approaches, as the information needed to answer questions can either be sparsely extracted from the entire video or found within a local region of the video. (Bottom) We propose the Continuous Perception Benchmark, a task that requires models to densely process input videos to answer questions correctly. We hope this task could facilitate the development of the next generation of vision models that emulate human ability to continuously perceive and process visual signals.

object permanence [20], as processing only a small number of frames may lead to learning superficial or spurious shortcut signals [21]. Additionally, such models could leverage the massive amount of available online video content for learning, which existing video models cannot do effectively due to excessive costs.

To facilitate the development of this envisioned next generation of vision models, we propose a new benchmark, called Continuous Perception Benchmark. This benchmark differs from existing video benchmarks [22, 23] by requiring models to continuously analyze the entire video stream for optimal performance (bottom of Figure 1). Most existing video benchmarks can often be tackled by analyzing just key frames [24, 25, 26, 11] or processing the video in segments [4, 22, 23]. However, the Continuous Perception Benchmark pushes models to develop a more comprehensive and uninterrupted understanding of the video. We evaluated several state-of-the-art foundational video models [15, 27, 11, 12, 15, 10, 11], both open-sourced and commercial, and found that none of them performed well on this newly proposed task. For instance, the best-performing model could only correctly answer 12% of the questions without any errors. This highlights the limitations of existing models and underscores the need for developing new techniques in this area.

## 2   Related Work

### 2.1   Multi-modal Foundational Models

The advent of multi-modal foundational models has marked a significant breakthrough in the field of artificial intelligence, enabling the integration of diverse data modalities such as text, images, and videos. In this paper we benchmark the open-sourced models and models with a public API: Video-ChatGPT [9], VideoLLaVa [10], LLoVi [15], PLLaVA [12], VideoChat2 [11], and Gemini [27]. Video-ChatGPT [9] computes spatiotemporal features from the videos by averaging frame-level features across temporal and spatial features, as input to the LLM through a learnable linear layer. VideoLLaVa [10] aligns images and videos before projection, enabling the LLM to learn from a unified visual representation. This process allows the LLM to comprehend both images and videos simultaneously. LLoVi [15] employs short-term visual captioners (such as LaViLa and BLIP2) to create textual descriptions for brief video segments. An LLM then compiles these detailed, short-term captions to perform the long-range reasoning necessary for LVQA. This approach enables LLoVI to effectively manage long-duration videos. PLLaVA [12] employs a simple pooling strategy to smooth

the feature distribution along the temporal dimension as input to the LLM. VideoChat2 [11] bridges LLM with a powerful vision foundational model [28], and trains the model on diverse instruction-tuning data with a novel progressive training paradigm. Gemini [27] is jointly trained across image, audio, video, and text data for the purpose of building a model with strong generalist capabilities across modalities.

## 2.2 Video Benchmarks

Various video benchmarks have been introduced over the years to advance video understanding technologies [25, 29, 30]. Early benchmarks focused on specific tasks such as activity classification [24, 25, 26], motion understanding [31], or movie analysis [32]. With the advent of visual foundation models [8, 6, 7, 9, 10], recent benchmarks have become more comprehensive, evaluating a wide range of model capabilities [33, 11] and often sourcing data from multiple existing video benchmarks [11, 34]. Another trend in benchmarking focuses on assessing long-form video understanding abilities [4, 22, 23]. Despite these diverse approaches, most existing benchmarks fall into two categories, where the information for answering the question can be extracted by either sparsely sampling several key frames [24, 25, 26, 11], or by captioning each small segments independently and then summarizing the resulting captions with language models [4, 22, 23]. Our proposed benchmark stands apart, as it requires the model to continuously process the entire input video. The information needed to answer the questions is densely distributed throughout the video, demanding continuous perception of visual stimuli as humans do.

## 2.3 Synthetic Datasets in Computer Vision

Our work, which involves synthetically generated data, is closely related to other research in computer vision. Their primary focus is to employ synthetic training data for real-world applications such as optical flow [35], point tracking [36], scene understanding [37, 38], and human pose understanding [39, 40, 41]. Another use of synthetic datasets is to investigate model capabilities in controlled environments. For spatial reasoning, some studies [42] render predefined objects using softwares like Blender [43]. More recently, research focusing on embodied agents has leveraged advanced simulators [44, 45] to create realistic environments. These simulators are equipped with a wide variety of assets and use physics engines like PyBullet [46] to generate more accurate and physically plausible scenes. This approach allows for a detailed examination of models' abilities in settings that closely mimic real-world conditions.

# 3 Continuous Perception Benchmark

To fill in the gap of existing benchmarks, Continuous Perception Benchmark (CPB) aims to build a video question and answering dataset that requires continuous processing of video frames. We use it to benchmark multi-modal foundational models to assess their capabilities for continuous perception.

## 3.1 Generation Method

We curate the dataset using OmniGibson [45] (MIT License), a simulation environment built upon NVIDIA's Omniverse platform. We select a 3D scene and populate it with furniture such as chairs and tables, then randomly place objects on the tables. Then videos are rendered with a moving camera following a specific trajectory (Figure 2). The task is simply asking how many of a specific objects are shown in the input video. Despite its simplicity, in the experiment section we show none of the existing state-of-the-art video models can perform well on the task.

The basic version of the dataset is created by having a camera move at a consistent speed across a room, maintaining a fixed direction to capture a panoramic view. This process results in a 20-second video at 30 fps for each instance. This method ensures that the visual data encompasses a continuous and seamless sweep of the entire room, providing comprehensive spatial context. To answer questions like

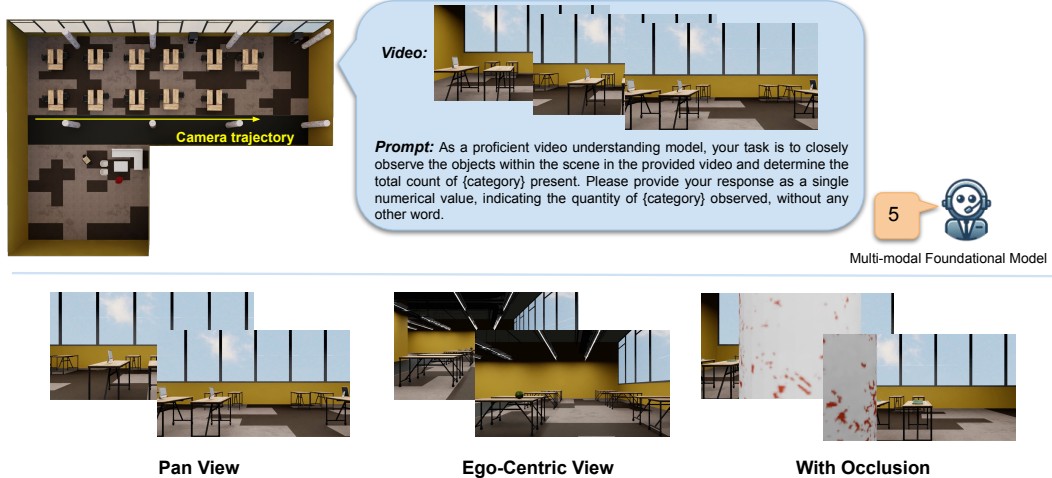

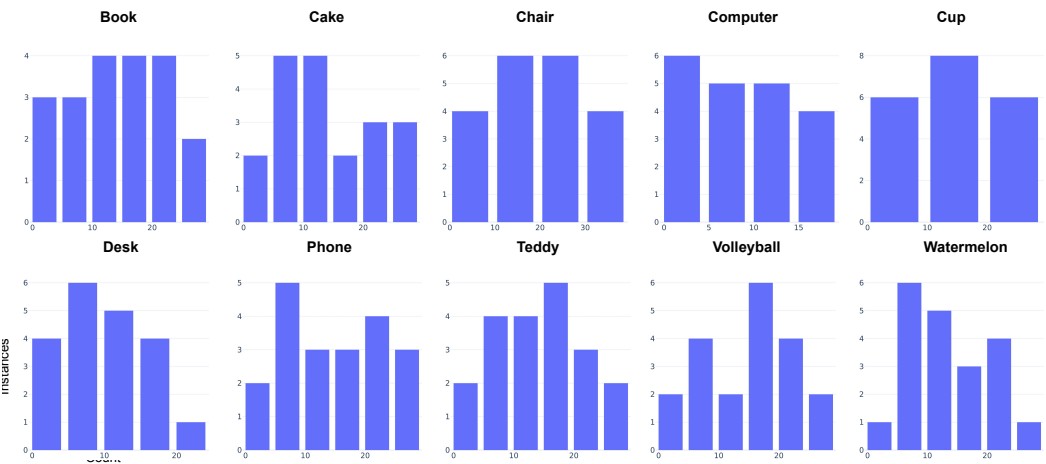

Figure 2: Top: Data generation (left) and benchmarking (right) illustration. Bottom: different variations of the benchmark.

Figure 3: Groundtruth count distribution for different target categories.

"how many desks are there in the room?", the model must thoroughly understand spatial relationships and environmental context, which requires processing the input video densely and continuously.

We select 10 object categories from the Behavior-1K database [45]: book, cake, chair, computer, cup, desk, phone, teddy bear, volleyball, and watermelon. For each category, we randomly sample 20 different scene configurations with different number of target object present at different locations, resulting a total of 200 test instances. Figure 3 shows the distributions of the ground truth count for different categories, which are roughly evenly represented across counts ranging from 1 to 30.

## 3.2 Evaluation Method

Following previous repetition counting works [47, 48, 49, 50], we use Mean Absolute Error (MAE), Root-Mean-Square-Error (RMSE), Off-By-One accuracy (OBO), Off-By-Zero (OBZ) as evaluation metrics, calculated as Eqs. 1 and 2 respectively. We additionally report Off-By-Five (OBF) accuracy (Eq. 3). The metrics OBF, OBO, and OBZ exhibit increasing levels of stringency for precise count accuracy. RMSE is more robust for evaluating diverse counts, as it is less biased towards smaller

counts compared to MAE.

$$MAE = \frac{1}{|\Omega|} \sum_{i \in \Omega} \frac{|c_i - \tilde{c}_i|}{c_i} \quad ; \quad RMSE = \sqrt{\frac{1}{|\Omega|} \sum_{i \in \Omega} (c_i - \tilde{c}_i)^2} \tag{1}$$

$$OBZ = \frac{1}{|\Omega|} \sum_{i \in \Omega} \mathbb{1}(|c_i - \tilde{c}_i| \le 0) \quad ; \quad OBO = \frac{1}{|\Omega|} \sum_{i \in \Omega} \mathbb{1}(|c_i - \tilde{c}_i| \le 1) \tag{2}$$

$$OBF = \frac{1}{|\Omega|} \sum_{i \in \Omega} \mathbb{1}(|c_i - \tilde{c}_i| \le 5) \tag{3}$$

$c_i$, $\tilde{c}_i$ are the ground-truth and predicted counts for $i_{th}$ video in the dataset $\Omega$. $\mathbb{1}$ is the indicator function.

# 4   Experiments

In this section, we will first introduce the various baseline models we evaluated on the proposed continuous perception benchmark. Then, we will present the experiment results and provide a detailed analysis of the model predictions.

## 4.1   Baselines

We evaluated several models aimed at video understanding. Specifically, Video-LLaVA [10], PLLaVA [12], VideoChat2 [11], and Video-ChatGPT [9] represent open-source multimodal models that generate answers directly from input video and question descriptions. LLoVi [15] represents models that first caption small, separate chunks of the input video, then summarize the captions of all chunks, and answer the question using a large language model (LLM). For commercial models, we evaluated Gemini [27] from Google. For all the open-source models, we utilized the inference code and released checkpoints from the official implementations. Figure 4 summarizes prompts used for different models, we used the 'VLM Prompt' for Video-LLaVA, PLLaVA, VideoChat2, Video-ChatGPT, and Gemini, and 'Captioning Prompt', 'LLM Prompt' for captioning part and answer generation part for LLoVi respectively. Note that we made small changes to the captioning prompt for LLoVi to deliberately instruct the captioning to output specific quantities of the target object. All open-source models are evaluated on an A6000 server.

**Video-LLaVA [10].**   Video-LLaVA represents a simple and robust multi-modal foundation model baseline where the visual representation is aligned with feature space of a large language model resulting in a unified large vision-language model. The model is trained on a mixed of image and video datasets where the image and video are first aligned before projecting to language feature space. It operates on input videos by sampling eight frames.

**PLLaVA [12].**   PLLaVA employs a simple pooling strategy to smooth the feature distribution along the temporal dimension as input to the LLM. This is shown to effectively reduce the dominant impacts from the extreme features. Our experiments were conducted using the 7B version of the model. When processing videos, PLLaVA samples 16 frames at a resolution of 336.

**VideoChat2 [11].**   VideoChat2 introduces a progressive training approach that incorporates a diverse range of multimodal instructions. This method effectively aligns video and language modalities. Our experiment utilized the 7B version of the model, processing input videos with 16 frames at a resolution of 224.

**Video-ChatGPT [9].**   Video-ChatGPT leverages CLIP-L/14 as the visual encoder to extract both spatial and temporal video features and the spatiotemporal features are computed through averaging frame-level features across temporal and spatial dimenions respectively. It sample the input video with 100 frames at resolution 224.

Figure 4: Prompts used for different models.

**LLoVi [15].** LLoVi is a framework designed for Long-Range Video Question Answering (LVQA). This method consists of two stages: initially, short-term visual captioners (such as LaViLa and BLIP2) generate textual descriptions for brief video segments spanning from 0.5 to 8 seconds. Subsequently, a Large Language Model (LLM) consolidates these short-term captions and conducts long-range reasoning. In our experiment, we employed BLIP2 for captioning and Llama-7B for summarizing the captions and answering the questions.

**Gemini [27].** Gemini is a family of highly capable multimodal models developed by Google. Gemini models are trained jointly across image, audio, video and text data for strong generalist capabilities across modalities. We tested Gemini-1.5-Flash and Gemini-1.5-Pro version on our proposed benchmark.

## 4.2 Experiment Results

Table 1 summarizes the overall evaluation results across different metrics. Notably, all models perform poorly on the proposed benchmark. Specifically, the best model, Gemini-1.5-Flash, correctly answers the questions only 12% of the time (OBZ). The predicted count is within one of the ground truth (OBO) only 20% of the time, and within five (OBF) 52% of the time. The mean absolute error (MAE) and root mean square error (RMSE) are also high, at 0.5 and 8.54, respectively. The performance of other open-source models is even worse, with OBO as low as 6% and RMSE as high as 14 (Video-LLaVA). This indicates that none of the existing video models can successfully complete the proposed task, which requires continuously modeling the entire input video and aggregating information perceived over time. Among the open-source models, LLoVi performs the best, with an OBF greater than 50% (compared to less than 45% for the others) and an RMSE lower than 9 (while others are higher than 11.5). This superior performance may be attributed to LLoVi's approach of dividing the input video into chunks and captioning each chunk, allowing it to process more input frames than the other models. Table 2 details the MAE for each object category. It shows that performance of different models varies across categories. For instance, LLoVi performs relatively better on 'watermelon' (0.28) than on 'cake' (0.44), while Gemini-1.5-Flash shows better performance on 'cake' (0.28) than on 'watermelon' (0.40).

**Distribution of predicted counts.** To further understand the models' predictions, we plot the distribution of predicted counts for each model, as shown in Figure 5. For Video-LLaVA and PLLaVA, most predicted counts are under 5, including cases where the model outputs a sentence without a valid number, which we set to 0. Video-ChatGPT's answers mostly fall under 2 and between 10-15. LLoVi predicts most answers under 20, while Gemini predicts most answers under 15. Most surprisingly, VideoChat2 almost always predicts counts within the 10-12 range. The striking disparity

Table 1: Overall results for different models.

| Model | OBZ | OBO | OBF | MAE | RMSE | CORR |
|---|---|---|---|---|---|---|
| Video-LLaVA | 0.01 | 0.06 | 0.23 | 0.87 | 14.07 | 0.43 |
| PLLaVA | 0.03 | 0.10 | 0.29 | 0.76 | 12.64 | 0.45 |
| VideoChat2 | 0.04 | 0.12 | 0.43 | 1.03 | 12.17 | 0.31 |
| Video-ChatGPT | 0.02 | 0.10 | 0.33 | 1.04 | 11.86 | 0.11 |
| LLoVi | 0.04 | 0.17 | 0.53 | 0.78 | 8.86 | 0.45 |
| Gemini-1.5-Flash | 0.12 | 0.20 | 0.52 | 0.50 | 8.54 | 0.72 |
| Gemini-1.5-Pro | 0.06 | 0.15 | 0.45 | 0.52 | 9.01 | 0.83 |

Table 2: Mean Absolute Error (MAE) of different models for all categories.

| Model | BO | CA | CH | CO | CU | DE | PH | TE | VO | WA | All |
|---|---|---|---|---|---|---|---|---|---|---|---|
| Video-LLaVA | 0.98 | 0.87 | 0.68 | 1.03 | 0.93 | 0.65 | 0.89 | 0.89 | 0.90 | 0.91 | 0.87 |
| PLLaVA | 1.00 | 0.79 | 0.54 | 0.79 | 0.90 | 0.45 | 0.95 | 0.59 | 0.82 | 0.77 | 0.76 |
| VideoChat2 | 1.29 | 0.88 | 1.39 | 1.25 | 1.08 | 1.39 | 0.89 | 0.99 | 0.62 | 0.54 | 1.03 |
| Video-ChatGPT | 1.42 | 0.83 | 1.09 | 1.01 | 1.25 | 0.68 | 1.35 | 1.01 | 0.61 | 1.16 | 1.04 |
| LLoVi | 0.87 | 0.44 | 0.95 | 1.60 | 1.08 | 0.76 | 1.06 | 0.30 | 0.47 | 0.28 | 0.78 |
| Gemini-1.5-Flash | 0.22 | 0.28 | 0.77 | 0.76 | 0.81 | 0.51 | 0.51 | 0.30 | 0.46 | 0.40 | 0.50 |
| Gemini-1.5-Pro | 0.45 | 0.39 | 0.76 | 0.72 | 0.49 | 0.38 | 0.60 | 0.38 | 0.55 | 0.45 | 0.52 |

between the predicted count distribution and the ground truth count distribution (shown on the left side of Figure 3) raises the question: "Does the model ever make predictions based on the input video?" To investigate this, we calculate the correlation between predicted counts and ground truth counts and summarize the results in the rightmost column of Table 1. The analysis reveals that, except for two Gemini models, which show a correlation of 0.72 and 0.83 for 1.5-Flash and 1.5-Pro respectively, all other models' predictions have a correlation with the ground truth of less than 0.5. This is the case despite LLoVi demonstrating similar performance to Gemini models on OBF and RMSE metrics.

**Distribution of correct predictions.** Figure 6 illustrates the percentage of correct predictions made by Gemini-1.5-Flash for each ground-truth count, as measured by OBZ, OBO, and OBF. The model demonstrates relatively better accuracy when the ground-truth count is low. However, when there are more than 8 target objects, the best OBO is less than 30%. This is understandable because higher ground-truth counts imply that objects are likely spread across different times rather than being

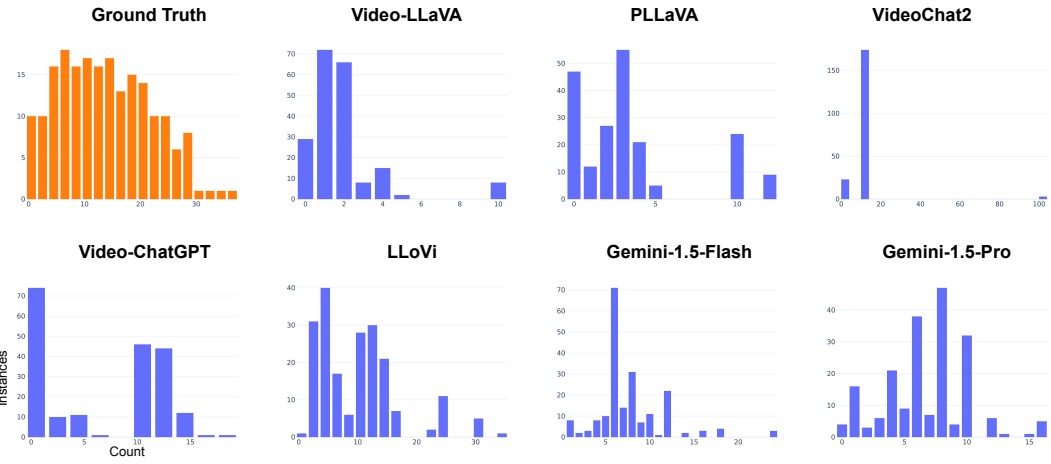

Figure 5: Predicted count distribution for different models.

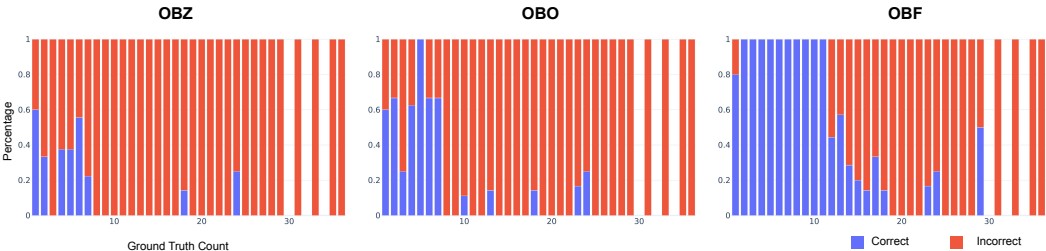

Figure 6: Distribution of correct prediction for Gemini-1.5-Flash. It shows that the model performs well when the ground truth count is low but struggles when there are more than 10 target objects in the scene.

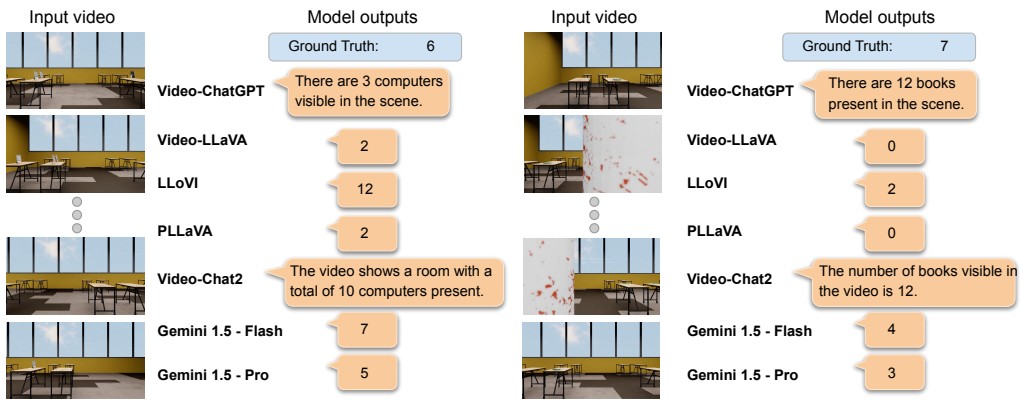

Figure 7: Examples from the proposed benchmark as well as the models' generated answer. Despite explicit instructions to output only a single number, some models still produce a complete sentence. When this occurs, we extract the first number from the output sentence as the model's prediction. If no number is present in the sentence, we set the prediction to zero.

concentrated in a local region. This situation requires the integration of a longer temporal context, which the model struggles to achieve effectively.

## 4.3 Additional Experiments

All experiments presented in the previous sections were conducted on the base version of the dataset, where the total length of the video is 20 seconds and the camera moves at a uniform speed. In this section, we conduct experiments with different variations of the base dataset. Table 3 summarizes the results of the Gemini-1.5-Flash model.

**With occlusion.** To simulate real-world scenarios where objects or structures can temporarily block the line of sight, we place pillars within the room. As the camera moves across the room, these pillars periodically obstruct the view, resulting in some frames being occluded (Bottom right of Figure 2). The occlusions challenge models to infer and reason about the environment despite partial visibility, testing their robustness and capability to handle incomplete or obstructed visual data. Despite the added difficulty, Gemini-1.5-Flash shows similar performance to the base version, indicating that additional occlusion does not influence the model's predictions.

**Nonuniform camera speed.** Furthermore, to explicitly discourage models from employing sparse uniform sampling, we introduce variations in the speed of the camera movement. Specifically, instead of using a uniform camera speed, we randomly sample from one of three movement patterns: starting fast and then slowing down, starting slow and then speeding up, or starting with a speedup followed

Table 3: Performance of Gemini-1.5-Flash on different variations of the dataset. 'Base' is the setting where the camera moves at a constant speed and captures a 20-second third-person view. 'Occlusion' introduces an additional foreground object, resulting in occlusion. 'Nonuniform' varies the camera speed. '5s Length' and '2min Length' are versions with total video lengths of 5 seconds and 2 minutes, respectively. "Egocentric" is the setting where the camera captures the first-person view. The model is not sensitive to foreground occlusion. It performs worse on the nonuniform 5s and 20s settings, but shows better results on the egocentric and nonuniform 2min settings.

| Model | OBZ | OBO | OBF | MAE | RMSE | CORR |
|---|---|---|---|---|---|---|
| Base | 0.12 | 0.20 | 0.52 | 0.50 | 8.54 | 0.72 |
| Occlusion | 0.10 | 0.21 | 0.52 | 0.50 | 8.48 | 0.77 |
| Nonuniform Speed | 0.09 | 0.17 | 0.48 | 0.51 | 8.92 | 0.75 |
| 5s Length | 0.04 | 0.11 | 0.37 | 0.65 | 10.70 | 0.74 |
| 2min Length | 0.10 | 0.23 | 0.59 | 0.45 | 7.59 | 0.75 |
| Egocentric | 0.10 | 0.27 | 0.64 | 0.54 | 6.16 | 0.70 |

by a slowdown. Compared to base version, Gemini performs slightly worse in this setting, with the OBF droppoing from 54% to 48%, and the RMSE increasing from 8.54 to 8.92.

**Video lengths.** The base version of the dataset has a fixed length of 20 seconds. We also experimented with two versions with different total lengths: one at 5 seconds and one at 2 minutes. Note that for both of the versions, the camera speed is not constant as in the 'nonuniform speed' version. Gemini shows a relatively large performance degradation on the 5-second version, with the OBF decreasing from 52% to 37% and the MAE increasing from 0.5 to 0.65. This might indicate that Gemini processes videos with a fixed frames-per-second rate, resulting in insufficient frame sampling for the 5-second dataset. For the 2-minute version, the model shows a slight decrease in performance in OBZ but improved performance in all other metrics.

**Egocentric view.** Finally, we created a variation of the dataset with an egocentric view instead of a third-person view, as this is common in many real-world applications such as home robots. On this dataset, Gemini shows improved OBF (from 52% to 64%) and RMSE (from 8.54 to 6.16). This could suggest that the model might have a better spatial understanding when processing an egocentric view compared to a third-person view.

## 5    Conclusion

In summary, we introduce a novel benchmark called the Continuous Perception Benchmark. The key distinction of this benchmark is that, to answer questions correctly, models must densely process the entire video, in contrast to existing benchmarks where sparse sampling or processing video in chunks is sufficient. Evaluation of multiple state-of-the-art video foundation models demonstrates that none of them excel at this task, indicating the need for new techniques. We hope this benchmark could facilitate developing the next generation of vision models that mimic human capabilities to continuously perceive and process visual stimuli. This advancement could be crucial for acquiring essential knowledge such as compositionality, intuitive physics, and object permanence.

**Limitations and future work.**    One limitation of the dataset is its synthetic nature, which may present challenges when transferring models from simulation to real-world scenarios. However, our experiments indicate that existing models struggle to handle even synthetic data effectively. Future work could consider collecting more real-world data to improve the diversity of the datasets.

**Potential negative societal impacts.**    This paper introduces a challenging task along with benchmarked performance of multi-modal foundational models, aiming to enhance the continuous perception capabilities of video foundational models. While we emphasize responsible use, we acknowledge the potential for these powerful video understanding models to be exploited for malicious purposes, such as unauthorized surveillance and automated profiling.

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

# A  Appendix

Due to space limit, we only included the performance of Gemini-1.5-Flash for variations of the benchmarks in Section 4.3. Below we present the full evaluation results for all models.

Table 4: Overall results for different models on occlusion version.

| Model | OBZ | OBO | OBF | MAE | RMSE | CORR |
|---|---|---|---|---|---|---|
| Video-LLaVA | 0.02 | 0.06 | 0.23 | 0.87 | 14.08 | 0.44 |
| PLLaVA | 0.01 | 0.10 | 0.30 | 0.76 | 12.60 | 0.46 |
| VideoChat2 | 0.04 | 0.12 | 0.41 | 1.29 | 11.63 | -0.04 |
| Video-ChatGPT | 0.02 | 0.07 | 0.30 | 1.02 | 12.41 | 0.08 |
| LLoVi | 0.07 | 0.22 | 0.52 | 0.74 | 8.74 | 0.51 |
| Gemini-1.5-Flash | 0.10 | 0.21 | 0.52 | 0.50 | 8.48 | 0.77 |
| Gemini-1.5-Pro | 0.06 | 0.16 | 0.45 | 0.55 | 9.45 | 0.85 |

Table 5: Overall results for different models on nonuniform version.

| Model | OBZ | OBO | OBF | MAE | RMSE | CORR |
|---|---|---|---|---|---|---|
| Video-LLaVA | 0.01 | 0.06 | 0.25 | 0.89 | 14.34 | 0.30 |
| PLLaVA | 0.01 | 0.07 | 0.27 | 0.79 | 13.07 | 0.45 |
| VideoChat2 | 0.04 | 0.11 | 0.43 | 1.04 | 9.13 | 0.19 |
| Video-ChatGPT | 0.04 | 0.09 | 0.30 | 1.09 | 12.38 | 0.09 |
| LLoVi | 0.05 | 0.16 | 0.53 | 0.69 | 9.06 | 0.50 |
| Gemini-1.5-Flash | 0.09 | 0.17 | 0.48 | 0.51 | 8.92 | 0.75 |
| Gemini-1.5-Pro | 0.06 | 0.15 | 0.45 | 0.54 | 9.43 | 0.81 |

Table 6: Overall results for different models on 5-second version.

| Model | OBZ | OBO | OBF | MAE | RMSE | CORR |
|---|---|---|---|---|---|---|
| Video-LLaVA | 0.01 | 0.08 | 0.23 | 0.90 | 14.24 | 0.27 |
| PLLaVA | 0.01 | 0.09 | 0.29 | 0.79 | 13.14 | 0.43 |
| VideoChat2 | 0.04 | 0.13 | 0.42 | 1.06 | 9.05 | 0.19 |
| Video-ChatGPT | 0.03 | 0.10 | 0.35 | 0.95 | 11.76 | 0.22 |
| LLoVi | 0.04 | 0.11 | 0.33 | 0.77 | 12.29 | 0.28 |
| Gemini-1.5-Flash | 0.04 | 0.11 | 0.37 | 0.65 | 10.70 | 0.74 |
| Gemini-1.5-Pro | 0.03 | 0.09 | 0.33 | 0.65 | 11.14 | 0.82 |

Table 7: Overall results for different models on 2-minute version.

| Model | OBZ | OBO | OBF | MAE | RMSE | CORR |
|---|---|---|---|---|---|---|
| Video-LLaVA | 0.01 | 0.07 | 0.23 | 0.86 | 14.05 | 0.40 |
| PLLaVA | 0.03 | 0.08 | 0.27 | 0.80 | 13.29 | 0.44 |
| VideoChat2 | 0.04 | 0.11 | 0.46 | 1.02 | 9.76 | 0.29 |
| Video-ChatGPT | 0.02 | 0.12 | 0.36 | 0.96 | 12.39 | 0.09 |
| LLoVi | 0.06 | 0.19 | 0.53 | 0.73 | 8.92 | 0.42 |
| Gemini-1.5-Flash | 0.10 | 0.23 | 0.59 | 0.45 | 7.59 | 0.75 |
| Gemini-1.5-Pro | 0.09 | 0.18 | 0.50 | 0.47 | 8.52 | 0.83 |

Table 8: Overall results for different models on egocentric version.

| Model | OBZ | OBO | OBF | MAE | RMSE | CORR |
|---|---|---|---|---|---|---|
| Video-LLaVA | 0.03 | 0.09 | 0.30 | 0.71 | 11.32 | 0.64 |
| PLLaVA | 0.05 | 0.16 | 0.48 | 0.62 | 9.45 | 0.49 |
| VideoChat2 | 0.06 | 0.14 | 0.46 | 1.06 | 8.32 | 0.25 |
| Video-ChatGPT | 0.03 | 0.10 | 0.33 | 0.99 | 12.87 | 0.00 |
| LLoVi | 0.06 | 0.14 | 0.47 | 0.98 | 9.84 | 0.31 |
| Gemini-1.5-Flash | 0.10 | 0.27 | 0.64 | 0.54 | 6.16 | 0.70 |
| Gemini-1.5-Pro | 0.09 | 0.22 | 0.50 | 0.42 | 8.52 | 0.82 |

