# Supplementary for Continuous Perception Benchmark

**Zeyu Wang, Zhenzhen Weng, Serena Yeung-Levy**
Stanford University
{wangzeyu,zzweng,syyeung}@stanford.edu

## 1 Dataset

### 1.1 Motivation

**For what purpose was the dataset created?**

The primary aim of the proposed dataset is to assess the video comprehension capabilities of multi-modal models. Unlike most existing datasets, this one emphasizes evaluating the model's continuous perception and processing of visual signals, simulating the way humans handle visual information.

**Who created this dataset (e.g., which team, research group) and on behalf of which entity (e.g., company, institution, organization)?**

The dataset was developed by a group of researchers at Stanford University.

### 1.2 Composition

**What do the instances that comprise the dataset represent (e.g., documents, photos, people, countries)?** Are there multiple types of instances (e.g., movies, users, and ratings; people and interactions between them; nodes and edges)? Please provide a description.

Each instance consists of a video, a question, and an answer. The video depicts an office scene featuring a varying number of objects.

**How many instances are there in total (of each type, if appropriate)?**

There are in total 200 video question-answer pairs.

**Does the dataset contain all possible instances or is it a sample (not necessarily random) of instances from a larger set?** If the dataset is a sample, then what is the larger set? Is the sample representative of the larger set (e.g., geographic coverage)? If so, please describe how this representativeness was validated/verified. If it is not representative of the larger set, please describe why not (e.g., to cover a more diverse range of instances, because instances were withheld or unavailable).

The dataset includes all instances we created for the benchmark. Being synthetic, it allows for the generation of additional instances.

**What data does each instance consist of? "Raw" data (e.g., unprocessed text or images) or features?**In either case, please provide a description.

Each instance is composed of a video, a question and an answer for the question.

**Is there a label or target associated with each instance?**If so, please provide a description.

Yes, there is a groundtruth label for each instance.

Submitted to the 38th Conference on Neural Information Processing Systems (NeurIPS 2024) Track on Datasets and Benchmarks. Do not distribute.

**Is any information missing from individual instances?** If so, please provide a description, explaining why this information is missing (e.g., because it was unavailable). This does not include intentionally removed information, but might include, e.g., redacted text.

No.

**Are relationships between individual instances made explicit (e.g., users' movie ratings, social network links)?** If so, please describe how these relationships are made explicit.

N/A.

**Are there recommended data splits (e.g., training, development/validation, testing)?** If so, please provide a description of these splits, explaining the rationale behind them.

All instances in the dataset are designated for testing.

**Are there any errors, sources of noise, or redundancies in the dataset?** If so, please provide a description.

No.

**Is the dataset self-contained, or does it link to or otherwise rely on external resources (e.g., websites, tweets, other datasets)?** If it links to or relies on external resources, a) are there guarantees that they will exist, and remain constant, over time; b) are there official archival versions of the complete dataset (i.e., including the external resources as they existed at the time the dataset was created); c) are there any restrictions (e.g., licenses, fees) associated with any of the external resources that might apply to a future user? Please provide descriptions of all external resources and any restrictions associated with them, as well as links or other access points, as appropriate.

The dataset is self-contained.

**Does the dataset contain data that might be considered confidential (e.g., data that is protected by legal privilege or by doctor-patient confidentiality, data that includes the content of individuals non-public communications)?** If so, please provide a description.

No.

**Does the dataset contain data that, if viewed directly, might be offensive, insulting, threatening, or might otherwise cause anxiety?** If so, please describe why.

No.

**Does the dataset relate to people?** If not, you may skip the remaining questions in this section.

No.

**Does the dataset identify any subpopulations (e.g., by age, gender)?** If so, please describe how these subpopulations are identified and provide a description of their respective distributions within the dataset.

N/A.

**Is it possible to identify individuals (i.e., one or more natural persons), either directly or indirectly (i.e., in combination with other data) from the dataset?** If so, please describe how.

N/A.

**Does the dataset contain data that might be considered sensitive in any way (e.g., data that reveals racial or ethnic origins, sexual orientations, religious beliefs, political opinions or union memberships, or locations; financial or health data; biometric or genetic data; forms of government identification, such as social security numbers; criminal history)?** If so, please provide a description.

N/A.

## 1.3 Collection Process

**How was the data associated with each instance acquired?** Was the data directly observable (e.g., raw text, movie ratings), reported by subjects (e.g., survey responses), or indirectly inferred/derived from other data (e.g., part-of-speech tags, model-based guesses for age or language)? If data was reported by subjects or indirectly inferred/derived from other data, was the data validated/verified? If so, please describe how.

It is synthetically generated using the OmniGibson environment.

**What mechanisms or procedures were used to collect the data (e.g., hardware apparatus or sensor, manual human curation, software program, software API)?** How were these mechanisms or procedures validated?

It is synthetically generated using the OmniGibson environment by rendering a predefined scene.

**If the dataset is a sample from a larger set, what was the sampling strategy (e.g., deterministic, probabilistic with specific sampling probabilities)?**

N/A.

**Who was involved in the data collection process (e.g., students, crowdworkers, contractors) and how were they compensated (e.g., how much were crowdworkers paid)?**

The researchers listed in the author list involved in the data creation process.

**Were any ethical review processes conducted (e.g., by an institutional review board)?** If so, please provide a description of these review processes, including the outcomes, as well as a link or other access point to any supporting documentation.

No.

**Does the dataset relate to people?** If not, you may skip the remaining questions in this section.

No.

**Did you collect the data from the individuals in question directly, or obtain it via third parties or other sources (e.g., websites)?**

N/A.

**Were the individuals in question notified about the data collection?** If so, please describe (or show with screenshots or other information) how notice was provided, and provide a link or other access point to, or otherwise reproduce, the exact language of the notification itself.

N/A.

**Did the individuals in question consent to the collection and use of their data?** If so, please describe (or show with screenshots or other information) how consent was requested and provided, and provide a link or other access point to, or otherwise reproduce, the exact language to which the individuals consented.

N/A.

**If consent was obtained, were the consenting individuals provided with a mechanism to revoke their consent in the future or for certain uses?** If so, please provide a description, as well as a link or other access point to the mechanism (if appropriate).

N/A.

**Has an analysis of the potential impact of the dataset and its use on data subjects (e.g., a data protection impact analysis) been conducted?** If so, please provide a description of this analysis, including the outcomes, as well as a link or other access point to any supporting documentation.

N/A.

### 1.4 Preprocessing/cleaning/labeling

**Was any preprocessing/cleaning/labeling of the data done (e.g., discretization or bucketing, tokenization, part-of-speech tagging, SIFT feature extraction, removal of instances, processing of missing values)?**If so, please provide a description. If not, you may skip the remainder of the questions in this section.

No.

**Was the "raw" data saved in addition to the preprocessed/cleaned/labeled data (e.g., to support unanticipated future uses)?**If so, please provide a link or other access point to the "raw" data.

N/A.

**Is the software used to preprocess/clean/label the instances available?**If so, please provide a link or other access point.

N/A.

### 1.5 Uses

**Has the dataset been used for any tasks already?**If so, please provide a description.

No.

**Is there a repository that links to any or all papers or systems that use the dataset?**If so, please provide a link or other access point.

N/A.

**What (other) tasks could the dataset be used for?**

In addition to evaluation, the dataset can also be used to train video understanding models.

**Is there anything about the composition of the dataset or the way it was collected and preprocessed/cleaned/labeled that might impact future uses?**For example, is there anything that a future user might need to know to avoid uses that could result in unfair treatment of individuals or groups (e.g., stereotyping, quality of service issues) or other undesirable harms (e.g., financial harms, legal risks) If so, please provide a description. Is there anything a future user could do to mitigate these undesirable harms?

No.

**Are there tasks for which the dataset should not be used?**If so, please provide a description.

N/A.

### 1.6 Distribution

**Will the dataset be distributed to third parties outside of the entity (e.g., company, institution, organization) on behalf of which the dataset was created?**If so, please provide a description.

The dataset is open to public.

**How will the dataset will be distributed (e.g., tarball on website, API, GitHub)**Does the dataset have a digital object identifier (DOI)?

The data will be distributed through Google Drive or other hosting services like HuggingFace. It can be accessed now at `https://drive.google.com/drive/folders/1gvX3JOXdO6CMdCSMJGhwoCgWs5wK-nXb?usp=sharing`.

**Will the dataset be distributed under a copyright or other intellectual property (IP) license, and/or under applicable terms of use (ToU)?**If so, please describe this license and/or ToU, and

provide a link or other access point to, or otherwise reproduce, any relevant licensing terms or ToU, as well as any fees associated with these restrictions.

The dataset will be distributed under the CC BY-SA 4.0 license.

**Have any third parties imposed IP-based or other restrictions on the data associated with the instances?**If so, please describe these restrictions, and provide a link or other access point to, or otherwise reproduce, any relevant licensing terms, as well as any fees associated with these restrictions.

No.

**Do any export controls or other regulatory restrictions apply to the dataset or to individual instances?**If so, please describe these restrictions, and provide a link or other access point to, or otherwise reproduce, any supporting documentation.

No.

### 1.7   Maintenance

**Who will be supporting/hosting/maintaining the dataset?**

The dataset will be maintained by the research team listed in the author list.

**How can the owner/curator/manager of the dataset be contacted (e.g., email address)?**

The creators can be contacted by the emails mentioned in the author list.

**Is there an erratum?**If so, please provide a link or other access point.

No.

**Will the dataset be updated (e.g., to correct labeling errors, add new instances, delete instances)?**If so, please describe how often, by whom, and how updates will be communicated to users (e.g., mailing list, GitHub)?

Yes, the dataset will be updated whenever necessary.

**If the dataset relates to people, are there applicable limits on the retention of the data associated with the instances (e.g., were individuals in question told that their data would be retained for a fixed period of time and then deleted)?**If so, please describe these limits and explain how they will be enforced.

N/A.

**Will older versions of the dataset continue to be supported/hosted/maintained?**If so, please describe how. If not, please describe how its obsolescence will be communicated to users.

Yes.

**If others want to extend/augment/build on/contribute to the dataset, is there a mechanism for them to do so?**If so, please provide a description. Will these contributions be validated/verified? If so, please describe how. If not, why not? Is there a process for communicating/distributing these contributions to other users? If so, please provide a description.

Such requests can be communicated to the authors via email.