# OpenReview forum: "Continuous Perception Benchmark"
_NeurIPS.cc/2024/Datasets_and_Benchmarks_Track — Submitted to NeurIPS 2024 Track Datasets and Benchmarks_

### Official Review · Reviewer_zVtE · 2024-07-21
**The topic is interesting but there are a few concerns about the benchmark including similarity to past work, use of synthetic data and poor results being confounded with limitations of VLMs unrelated to continuous perception.**

**Rating:** 6
**Confidence:** 4

**Review:**

The paper proposes to tackle a very interesting problem of continuous perception. I wholly agree with the paper that next generation models need to process visual input continuously like humans.

The paper is largely well written and the motivation of the paper is clearly communicated.

However, there are few problems with the current instantiation of the benchmark,
1. How is it different from the EM-EQA benchmark from the OpenEQA [1] paper? The idea is basically the same: the agent is shown a video and asked a specific question about an object occurring in the video. The paper does not make any claims about “continuous” perception but I think it inherently proposes the same challenge.

2. Does the benchmark really need "continuous perception"? If we are somehow able to subsample the videos densely enough so that they include all the instances of the objects while fitting in the context window of these models, then that would be enough. The benchmark does not inherently stop you from subsampling.
Since the data is synthetically generated, it can be easily validated how much subsampling can we do, before it becomes impossible to correctly solve the benchmark.
Also, a better instantiation of the task would be to have a streaming video and then continuously ask the model the running count at random timestamps. This would ensure that a fixed number of frames will not cut it.


3. Also counting may not be the best task to test these models as these models are known to perform poorly on counting / other mathematical tasks. A nice task that I liked from the Gemini tech report is the needle in the haystack. I could see that sort of questions make a better benchmark.


4. The benchmark is based on synthetic data while these models are trained on largely real world images. I would have liked to see baseline classification performance of these models on the classification/detection of the selected objects in these videos. That would ensure that the problem is with “continuous perception” and not perception in general.

Given the above concerns, I can not suggest the acceptance of this paper in the current form.

[1] Majumdar, A., Ajay, A., Zhang, X., Putta, P., Yenamandra, S., Henaff, M., Silwal, S., Mcvay, P., Maksymets, O., Arnaud, S. and Yadav, K., 2024. Openeqa: Embodied question answering in the era of foundation models. In Proceedings of the IEEE/CVF Conference on Computer Vision and Pattern Recognition (pp. 16488-16498).

**Updated the score after rebuttal. For more information, check the discussion below.**

**Strengths:**

1. The paper aims to tackle a very important and relevant task. I fully agree that “continuous perception” is the next frontier for the SOTA VLMs.
3. The paper is largely well written and easy to understand.
3. The paper provides results for almost all the most relevant VLMs including both open-source and commercial models showing that all of them struggle with the task.

**Additional Feedback:**

No additional comments.

**Clarity:**

The paper is largely well written and easy to understand.
The only point of feedback would be to improve Figure 6. I’m not sure if the red bars are needed at all.

**Correctness:**

Yes, the submission is mostly correct.

My only concern is the use of synthetic data to establish this benchmark as most of these models are trained on largely real-world image databases. This raises the question if the poor performance of these models is due to them struggling to perform “continuous perception” or due to the Sim2Real gap.

**Documentation:**

The paper does not include a link for the dataset generated which is used in the benchmark.

**Limitations:**

The authors include a limitations and societal impact section that sufficiently addresses the limitations/concerns.

**Opportunities For Improvement:**

See the review section for weaknesses / opportunities for improvement.

**Relation To Prior Work:**

There have been quite a bit of work recently where the video understanding benchmarks are going beyond simple VQA where the model is required to do continuous reasoning in an online manner to be able to perform well which this paper fails to mention and discuss. Some of the works are,
1. Majumdar, Arjun, et al. "Openeqa: Embodied question answering in the era of foundation models." Proceedings of the IEEE/CVF Conference on Computer Vision and Pattern Recognition. 2024.
2. Wang, Xin, et al. "Holoassist: an egocentric human interaction dataset for interactive ai assistants in the real world." Proceedings of the IEEE/CVF International Conference on Computer Vision. 2023.

**Summary And Contributions:**

The paper proposes a benchmark that evaluates the ability of visual models to process visual inputs continuously. In effect, the benchmark is a video question answering task where the questions are designed in a way that they can not be answered in a typical video understanding setting where the model is fed a sub-sampled video.

The authors generate 200 videos in an automatically populated scene with a moving camera. They use counting instances of a category as the main task i.e. each video is paired with a "how many x in this room" question. The paper benchmarks 7 state-of-the-art open source and commercial VLMs on the benchmark and shows that all the models perform poorly showing both the need for such a benchmark and also new advancements required to solve it.

---

> ### Author Rebuttal · Authors · 2024-08-17
>
> We thank the review for the time, feedback and the appreciation of the core statement of our work. Please kindly refer to the following for the concerns raised:
>
> **Q: Difference with dataset like OpenEQA.**
>
> **A:** We want to clarify that a central feature of our continuous perception benchmark is that the information necessary to answer the question is distributed throughout the entire video, requiring the core video-language model to process all frames. In contrast, datasets like OpenEQA do not have this requirement; instead, the relevant information is localized within specific regions of the video. For instance, in a typical OpenEQA question like "I can’t find my keys, where did I leave them?", a single target frame (or sometimes a few frames) is sufficient to answer the question. To address such questions, one approach is to first use a multi-modal model to identify relevant frames (containing keys), and the core video-language model then only needs to process these "sparsely sampled" frames to provide an answer. Therefore, datasets like OpenEQA do not necessitate continuous perception as our benchmark does.
>
> **Q: Does the task need continuous perception.**
>
> **A:** We would like to clarify that the continuous perception means that the sampling rate is sufficient enough to capture the continuous nature of the scene, instead of strictly processing every frame of the input video. To make an extreme case, we could generate a video traversing the same scene in ten days (really really slow camera movement), then both subsampling 100x, 1000x, would still be considered continuous perception. Therefore "subsample the videos densely enough so that they include all the instances of the objects" is in this sense continuous perception.
>
> **Q: Comparison to needle in the haystack task.**
>
> **A:** We appreciate the reviewer's feedback and their recognition of the "needle in the haystack" task, which we also find very valuable. Despite it, we would like to emphasize that our continuous perception benchmark evaluates a different dimension of performance. While the needle in the haystack task assesses a model’s ability to retrieve information from a long context where the target information is localized, our benchmark evaluates a model’s capacity to integrate information over time, where the target information is distributed. This temporal aggregation is crucial for understanding the temporal continuity in "video", rather than merely handling "multiple images". Consequently, despite models like Gemini being able to sample continuously due to their long context capabilities, they don't "perceive continuously", as evident by low performance on our proposed benchmark.
>
> **Q: Results might be confounded with out-of-domain data.**
>
> **A:** To investigate whether the error arises from the need for continuous perception, we conducted an experiment where the camera remained fixed throughout the video, creating a static scene. In this setup, we asked the model to count the number of visible target objects. We tested this with "watermelon" and the Gemini-1.5-Flash model.
>
> This is the result on the static scene,
>
> | OBZ | OBO | OBF | MAE | RMSE | CORR |
> | ----- | ------ | ----- | ----- | -------- | ------- |
> |  0.85 | 1.00 | 1.00 |	0.05 | 0.39 | 0.97 |
>
> in comparison to the result on the proposed continuous perception benchmark,
>
> | OBZ | OBO | OBF | MAE | RMSE | CORR |
> | ----- | ------ | ----- | ----- | -------- | ------- |
> | 0.15 | 0.20 | 0.45 | 0.40 | 8.31 | 0.87 |
>
> This result shows that the proposed continuous perception task poses unique "continuous perception" challenge besides the "perception" challenge.
>
> **Q: Access to data.**
>
> **A:** Sorry for the confusion, the link to data was included on Ln. 153 in the appendix. Here is the [link](https://drive.google.com/drive/u/2/folders/1gvX3JOXd06CMdCSMJGhwoCgWs5wK-nXb) again.

---

> > ### Comment · Reviewer_zVtE · 2024-08-28
> >
> > I thank the authors for their rebuttal. I'm still not quite convinced,
> >
> > **Difference with OpenEQA.**
> >
> > The authors here describe an instantiation of a model that might be able to solve OpenEQA question where there is a "lightweight" multi-modal model that performs a first pass and then a video model takes over and uses the output from this "lightweight" model to process only a small portion of the video to solve the task. However, the system as whole still have to see the WHOLE video to answer the question -- the need to distinguish between an image model doing a first pass and video model doing a second pass is kind of arbitrary in my opinion. It is perfectly reasonable to assume that a video model actually end up doing something similar under-the-hood to solve both OpenEQA and continuous perception benchmark. (An aside, maybe we humans already do such processing RE. System 1 and System 2 modes of thinking)
> >
> >  **Does it need continuous perception?**
> >
> > Using this rationale, can't we come up with a method in which an image model first looks at the video and selects the frames to be viewed by the video model? For example, an image model could be shown Tth image (T+1)th image and instructed such that if all the "watermelons" appearing in image T also appear in image (T+1), then discard image T. Does it not contradict the authors' previous claim in the answer of OpenEQA question?
> >
> > **Comparison with needle in the haystack task**
> >
> > My question was about the use of counting task in perception benchmark with the main concern being that as the number increase, the models capabilities to predict correct number generally goes down quite a LOT! In other words, we know these models suck at counting -- why use a task we know they suck at to understand another capability? We will only confound results.
> >
> > **New Results**
> >
> > After looking at the data (and specially watermelon videos), I find it highly surprising that the Gemini model is able to perform at such a high level in the static case. In most "high count" case, it is almost impossible to determine the number of watermelons from a static frame (at least the starting view) due to occlusions. I personally went through most of the videos and definitely made mistakes in more than 3 (15% of 20) of those videos.
> >
> > Maybe I'm missing some details. Most importantly which view of the camera was chosen to be fed into Gemini?

---

> > ### Author Rebuttal · Authors · 2024-08-29
> >
> > We thank the reviewer for the additional feedback. Please kindly refer to the following as the response:
> >
> > **1. Clarification on Continuous Perception**
> >
> > We want to clarify that "continuous perception" is not merely about "processing all frames." Instead, it involves **processing all frames collectively to achieve a holistic understanding of the continuity of space and time**. In this context, a two-step system—where an image-language model extracts key frames and a VLM generates answers based on those key frames—is not considered continuous perception. Although this system processes all frames, the individual models only handle a subset at a time (the first model processes one image at a time, and the second model deals with only a few key frames), preventing a holistic grasp of spatial and temporal continuity. Similarly, benchmarks like OpenEQA differ from the proposed continuous perception benchmark because answering questions in OpenEQA does not necessitate continuous perception, and the described two-step system is sufficient for that task.
> >
> > The described two-step system is unable to solve the proposed task because the sampling process in the first step disrupts the spatial-temporal continuity for the second step. Consequently, the second step would fail to establish correct correspondences. Since the frames contain overlapping objects (e.g. frame 1 contains objects 1,2,3, frame 2 contains 2,3,4), this disruption can result in the same object being counted multiple times.
> >
> > **2. Clarification on additional experiment with static camera**
> >
> > Sorry for any confusion. To clarify, the videos used for the static camera experiment are not directly from the proposed continuous benchmark. This experiment was designed to eliminate potential confounding errors from domain gaps in synthetic images. Therefore, we generated new videos for the same counting task using the same pipeline but without requiring continuous perception—i.e. all objects present are visible within the static camera frame. Gemini’s high performance in this setup shows that it can recognize the objects effectively. Consequently, any errors in the proposed benchmark arise from its requirement for a holistic understanding of spatial and temporal continuity.
> >
> > **3. Issue with counting**
> >
> > Apologies for the initial misunderstanding. We generally agree with the reviewer that excessive counting can indeed challenge existing models. However, as Figure 6 shows, even with fewer than 10 objects, the Gemini model's performance remains notably low, with an OBZ of around 20%. This contrasts sharply with the over 80% OBZ achieved in the static camera experiment (object count ranging from 1-10). Therefore, we would argue that the nature of the counting task does not significantly undermine the main point of the paper.

---

> > > ### Comment · Reviewer_zVtE · 2024-08-29
> > >
> > > I thank the authors for their reply.
> > >
> > > **1. On Continuous Perception.**
> > >
> > > I understand the need for "holistic understanding of the continuity of space and time" for continuous perception, but I still feel the insistence that a two-step system does not have such understanding is arbitrary -- as long as the system is able to solve the task, why not?
> > >
> > > Having said that, I believe we digress from the main point which was: how it is different from OpenEQA? And maybe the bottom-line should be, in OpenEQA the information required to answer any question is limited to a few frames as opposed to this benchmark where it is distributed along the whole video.
> > >
> > > **2. Clarification on additional experiment with static camera.**
> > >
> > > Thanks for the clarification. It would be great if such an experiment (maybe also updated to include all the classes) could be added to the main paper too.
> > >
> > > Overall, after a bit of back and forth with the authors, I am a bit more positive about the paper and will update the score to borderline accept 6 (from reject 4) while noting that there are still a few aspects of the paper that warrant a revisit (e.g. use of counting as a task and more complete experiments with the static camera)

---

> > > > ### Author Rebuttal · Authors · 2024-08-30
> > > >
> > > > We would like to thank the reviewer once again for their efforts and valuable feedback. We will include the static camera experiment in the main paper as recommended. Thank you.

---

### Official Review · Reviewer_Ric1 · 2024-07-23
**Continuous Perception Benchmark contributes towards understanding long term video data**

**Rating:** 6
**Confidence:** 4
**Clarity:** yes

**Review:**

The paper is well written and very easy to understand. The motivation for the problem was explained properly. The data generation process was explined. However I believe the authors can make some edits and add further details on the process. Please find the limitation section for further details on this part.
The evaluation process for benchmarking on the dataset with various VLM models were explained properly. The results were presented in a coherent manner and were very easy to follow from the tables and figures. I found it very interesting that even with all the training and the success of the Large language models, the state of the art VLMs fail to infer simple tasks like counting the no of computers in a room from video data. This shows that when it comes infering from spatio-temporal consistencies, we still have long way to go and opens up interesting avenues to explore for future.

My concern is why did models like Gemini or Video-ChatGPT performe so poorly on this dataset with such low correlation index (Table 1). Is it because these models have never seen this kind of data or trained to make such inferences?

**Strengths:**

Well written paper with a problem statement along with the open sourced dataset that is of relevence to the AI community. The evaluation technices were properly explained and were consistent. Even though this video dataset is a very simple and generated in a simulated env, it shows very interesting results when benchmarked on SOTA VLM model. The results opens up interesting avenues for future research for understanding scene from video and spatio-temporal consistecy.

**Additional Feedback:**

none

**Correctness:**

yes. To my knowledge the evaluation and experimental designs are done appropriately.

**Documentation:**

Dataset is open-sourced but documentation can be improved. Please follow comments on limitations.

**Limitations:**

Please follow my suggestions in Opportunities For Improvement.

**Opportunities For Improvement:**

The authors can provide some more details on the data generation process like,

What is a velocity of the camera when traversing in a room?

What are the velocity profiles for the camera when speed was variable as mentioned in line 212 on sec  Additional Experiments?
having a github page with the the code for creating the training data-pipeline for the dataset will be very useful.

As mentioned by the authors also, this is a video dataset in simulated environment which might not co-relate well with natural data.

**Relation To Prior Work:**

yes.

**Summary And Contributions:**

The authors created a small video perception dataset and benchmark for continuous video understanding. These dataset sheds light on some of the major short-comings of the current SOTA large vision language models like Gemini and Video-GPT when it comes to understanding an inferring from video data. The dataset is open sourced for public use.

---

> ### Author Rebuttal · Authors · 2024-08-17
>
> We thank the review for the time, feedback and the appreciation of our work. Please kindly refer to the following for the concerns raised:
>
> **Q: Velocity of the camera.**
>
> **A:** For the base version of the dataset (uniform speed, 20s length), the camera travels 25m, which equals a speed of 1.25m/s. For the version with nonuniform speed, we randomly select from three settings, gradually increasing, gradually decreasing or first increasing and then decreasing. The start/end speed is zero and peak speed is 2.5m/s. We appreciate the reviewer's suggestion to have a github repo for the data generation code and we will implement this recommendation.
>
> **Q: Correlation with real-world data.**
>
> **A:** While we acknowledge the concern, the primary goal of our benchmark is to foster the development of next-generation models capable of continuous video processing. Such models will inherently be computationally effective and capable of being trained on large volumes of real-world online data. Additionally, we intend for the dataset to serve solely as a test set for evaluating general vision-language models trained on web datasets, rather than for models specifically trained on synthetic data.

---

> > ### Comment · Reviewer_Ric1 · 2024-08-29
> >
> > Thank you for the clarifications. That clears my doubts.

---

### Official Review · Reviewer_L2mZ · 2024-07-24
**Continuous perception benchmark**

**Rating:** 4
**Confidence:** 3
**Correctness:** Seems correct.
**Clarity:** Clear enough.

**Review:**

As it stands, I will not support this paper to be accepted. The task seems too simplistic, and there is no support over the claim that the task is 1) significant enough, and 2) help evaluate the different models on other tasks. It also fails to convince me a true continuous model is needed. The scope seems too small for NeurIPS.

**Strengths:**

Exposition of the problem is clear enough. I just don't really buy it.

**Additional Feedback:**

Please comment on why the task is significant. Just because all models didn't work well on it does make it automatically significant on evaluating an existing model.

**Documentation:**

Seems okay.

**Ethics:**

No ethics issues detected.

**Limitations:**

Explained in the last section already.

**Opportunities For Improvement:**

- title seems overclaiming. There's only one task in the dataset and it is to count the number of objects.
- I am not sure if the counting requires continuous stream of video. You can still chunk it as long as it is reasonable compared to how fast the camera is panning. One can argue the model can accumulate knowledge over keyframes or chunks to achieve similar performance over model that truly take in continuous information.
- The author also failed to convince me why the counting problem is important for any downstream applications.
- I think hardware is one big concern for these keyframe-based or chunk-based approaches. Completely continuous stream is ideal but not possible.

**Relation To Prior Work:**

Seems okay.

**Summary And Contributions:**

This paper introduces a synthetic dataset of panoramic videos and a single task of asking video models to count number of objects in the video. They show that current models do not do well on this specific task.

---

> ### Author Rebuttal · Authors · 2024-08-16
>
> We thank the review for the time and feedback. Please kindly refer to the following for the concerns raised:
>
> **Q: The proposed task doesn't seem to need continuous processing.**
>
> **A:** We believe to successfully solve the proposed task, a continuous processing of input video is needed. For sparse sampling, it would miss part of the scene and corresponding objects (note that a sufficient sampling rate would be the continuous processing). For processing videos in chunks and then summarize, because of continuous nature of the space, it will result in repetitive detections for consecutive chunks. Thus, continuous processing ensures that all relevant information is captured without missing or repeating detections.
>
> **Q: The proposed task doesn't help evaluate different models on other tasks.**
>
> **A:** We believe that a general visual-language model capable of processing videos continuously and performing well on our proposed task would likely generalize effectively to other video benchmarks. However, since such a model does not yet exist, this prediction will need to be validated in the future.
>
> Additionally, in our evaluations, Gemini has demonstrated superior performance compared to other models, including open-source models like LLoVi and PLLaVA, which also show better correlation than their peers. This aligns with our understanding of the models' relative capabilities.
>
> Therefore, from both perspectives, we couldn't see the concern raised is a strong argument against the proposed task.
>
> **Q: Hardware is one big concern for these keyframe-based or chunk-based approaches. Completely continuous stream is ideal but not possible.**
>
> **A:** We acknowledge and agree that continuous processing of input videos presents significant computational challenges for current video models. However, we don't view this as a drawback of the proposed benchmark. On the contrary, the benchmark aims to encourage the development of new models that can process videos more efficiently. We believe that creating computationally efficient models will be crucial for utilizing the vast amount of online video data for training and achieving human-level visual capabilities.

---

### Official Review · Reviewer_CGQb · 2024-07-25
**Review for paper submission #1281**

**Rating:** 4
**Confidence:** 4
**Correctness:** The considered aspect is limited
**Clarity:** The paper describes the idea and meth…

**Review:**

This paper presents a video data generation pipeline based on 3D engine. The video is generated by mainly controling the camera movements. The generated data is then used for evaluating the large multimodal models.

**Strengths:**

- This paper considers the key challenges in Video LLM evaluation, particularly the capability to understand the video by using the entire video instead of sparsing sampled frames.
- This paper conducted a large number of experiments and compared the performance of several well-known models such as Gemini-1.5 and Video-ChatGPT, etc.

**Additional Feedback:**

N/A

**Documentation:**

N/A

**Opportunities For Improvement:**

The main concern is how to design questions that really need to process the entire video? Given the example shown in Fig 2, sparsely sampled frames can also be useful to answer the question of `how many desks are there in the room`. Currently, this paper only considers the counting task for evaluation, which is not convincing at all. Instead of counting, this paper ignores the key factors including the event continuity, causal relationship, and motion/action, which are natively the key features and are difficult to infer via sparsely sampled frames. As the paper only considers counting, the evaluation and contribution are reletively weak.

**Relation To Prior Work:**

There are discussion and comparison with prior works.

**Summary And Contributions:**

This paper introduces a video benchmark. The key characteristic of the proposed dataset is that to answer the provided questions correctly, by design the candidate models need to process the whole video.

---

> ### Author Rebuttal · Authors · 2024-08-16
>
> We thank the review for the time and feedback. Please kindly refer to the following for the concerns raised:
>
> **Q: Does the proposed task need continuous perception to solve?**
>
> **A:** We believe to successfully solve the proposed task, a continuous processing of input video is needed. For sparse sampling, it would miss part of the scene and corresponding objects (note that a sufficient sampling rate would be the continuous processing). For processing videos in chunks and then summarize, because of continuous nature of the space, it will result in repetitive detections for consecutive chunks. Thus, continuous processing ensures that all relevant information is captured without missing or repeating detections.
>
> **Q: Ignore key factors like event continuity, causal relationship, motion/action.**
>
> **A:** First, the proposed task arguably has the event continuity nature, i.e. the viewer is gradually observing the whole scene, one part at a time. Second, for casual relationship, motion/action, although they require temporal aspect (more than one frame), we found for many practical events related to these (as presented in many existing video benchmarks), they can be solved with sparse sampling. For example, for casual relationship "a push causing a bottle fall from table", only initial frame "hand contacting bottle" and last frame "bottle on ground" is needed. For motion/action "moving pen from left to right", only initial frame "pen on left" and last frame "pen on right" is needed. Therefore, we focus on the spatial counting task, which genuinely requires continuous perception in the way we understand it currently.

---

> > ### Comment · Reviewer_CGQb · 2024-08-31
> > **comments**
> >
> > While we appreciate the authors' response and understand their perspective, we still find the explanation insufficient in addressing the core concerns.
> >
> > Event Continuity: The authors mention that the task inherently involves event continuity because the viewer is gradually observing the whole scene, one part at a time. However, this interpretation seems limited. Event continuity in video analysis often implies understanding the flow of events over time, not merely the sequential observation of static parts of a scene. There’s a distinction between spatial observation and temporal event continuity, where the latter involves understanding the progression of events. The proposed task seems to be missing the temporal understanding discussed here.
> >
> > Causal Relationships: The authors argue that causal relationships can be inferred from sparse sampling by considering only the initial and final states of an event. While sparse sampling might capture the beginning and end states, it overlooks the intermediate states that often carry significant causal information. In the example provided ("a push causing a bottle to fall"), the act of pushing, the dynamics of the bottle tipping, and other contextual cues are crucial for a complete understanding of the causal relationship. The simplified approach risks missing out on important aspects that occur between the initial and final frames.
> >
> > Motion/Action Understanding: Similar to causal relationships, understanding motion or actions usually involves more than just the initial and final positions of objects. The intermediate frames often contain critical information about the nature and quality of the movement. For example, the speed, acceleration, and trajectory of the "pen moving from left to right" are all part of what defines the action, and these cannot be fully captured with just two frames.

---

> > ### Author Rebuttal · Authors · 2024-08-31
> >
> > We thank the reviewer for the additional feedback. Please kindly refer to the following as the response:
> >
> > **1. Clarification on event continuity, causal relationships and motion/action**
> >
> > We apologize for any confusion. To clarify, our argument is not that only two frames (initial and final) are sufficient to understand event continuity, causal relationships and motion/action. On the contrary, we fully agree with the reviewer that all frames are necessary for a comprehensive understanding of event continuity, causal relationships and motion/action.
> >
> > Our concern is that existing benchmarks for event continuity, causal relationships and motion/action—such as recognizing movement from left to right or right to left—can often be addressed by sparsely sampling a few frames (as is common with many video models). This approach may lead models to rely on shortcuts rather than truly learning about event continuity, causal relationships and motion/action.
> >
> > To encourage the development of models that genuinely understand event continuity, causal relationships and motion/action, we propose the continuous perception benchmark. This benchmark is designed with the principle that sparse sampling of frames is insufficient for solving the task. We hope this benchmark will drive the creation of models that process all frames, leading to a deeper and more accurate understanding of event continuity, causal relationships and motion/action compared to current models.
> >
> > **2. Clarification on counting task**
> >
> > As our main goal is about designing a benchmark that defies sparsely sampling models and facilitates development of models that holistically process all frames, we opted for the spatial counting task due to its simplicity and effectiveness in achieving this goal. Also note that the benchmark only contains a test set, and is intended to evaluate foundational visual models trained on large and diverse data sources. Therefore, we would argue that the specifics of the counting task does not significantly undermine the main argument of the paper.

---

> > > ### Comment · Reviewer_CGQb · 2024-08-31
> > > **comments**
> > >
> > > In my point of view, the use of the counting task remains insufficient to address all the challenges of continuous perception. Adding more tasks would greatly enhance the contribution of this paper.
> > >
> > > After careful consideration, I decide to keep my rating at it.

---

> > ### Author Response · Authors · 2024-09-01
> >
> > Although we respectfully disagree with the reviewer and believe that the specific form of the task does not substantially detract from the core aim of our paper—namely, to draw attention to this new area and foster the development of next-generation visual models that emulate continuous human perception—we still appreciate the reviewer's time and effort. Thank you.

---

### Decision · Program_Chairs · 2024-09-26

**Decision:**

Reject

**Comment:**

This paper obtained two rejects and two borderline accepts. While the paper proposes to tackle an interesting problem of continuous perception, the reviews raised concerns about the limited novelty, unconvincing justification, and missing details/experiments. The main concern was centered around the counting problem, which was the authors’ focus on continuous perception; is the counting problem adequate and critical for continuous perception? Despite the authors’ rebuttal and discussion, the reviewers remain unconvinced. AC also agrees with the reviewers that this submission ignores the key factors of continuous perception, such as event continuity, causal relationship, and motion/action. AC thus recommends rejecting this submission and encourages the authors to improve their paper based on the reviewers’ comments and submit it to another venue.